# Finite-Time Thermodynamic Modeling and Optimization of Short-Chain Hydrocarbon Polymerization-Catalyzed Synthetic Fuel Process

**DOI:** 10.3390/e24111658

**Published:** 2022-11-15

**Authors:** Yajie Yu, Shaojun Xia, Qinglong Jin, Lei Rong

**Affiliations:** 1College of Power Engineering, Naval University of Engineering, Wuhan 430033, China; 2College of Nuclear Science and Technology, Naval University of Engineering, Wuhan 430033, China

**Keywords:** synthetic fuel, chemical process, entropy generation rate, finite-time thermodynamics

## Abstract

The short-chain hydrocarbon polymerization-catalyzed synthetic fuel technology has great development potential in the fields of energy storage and renewable energy. Modeling and optimization of a short-chain hydrocarbon polymerization-catalyzed synthetic fuel process involving mixers, compressors, heat exchangers, reactors, and separators are performed through finite-time thermodynamics. Under the given conditions of the heat source temperature of the heat exchanger and the reactor, the optimal performance of the process is solved by taking the mole fraction of components, pressure, and molar flow as the optimization variables, and taking the minimum entropy generation rate (MEGR) of the process as the optimization objective. The results show that the entropy generation rate of the optimized reaction process is reduced by 48.81% compared to the reference process; among them, the component mole fraction is the most obvious optimization variable. The research results have certain theoretical guiding significance for the selection of the operation parameters of the short-chain hydrocarbon polymerization-catalyzed synthetic fuel process.

## 1. Introduction

Short-chain hydrocarbon polymerization-catalyzed synthetic fuel technology [1,2] is a technology that uses seawater as raw material to convert nuclear or renewable energy into chemical energy. The technology includes two-step reactions of CO_2_ hydrogenation to light olefins (CHLO) and olefin oligomerization. Firstly, H_2_ and CO_2_ are obtained from seawater by electrochemical acidification as raw materials. Secondly, C_2_-C_4_ unsaturated hydrocarbons are obtained by CHLO. Thirdly, C_2_-C_4_ unsaturated hydrocarbons are further synthesized into C_9+_ liquid fuel by means of olefin oligomerization. This technical path has not yet entered the stage of engineering application and there are many problems, such as high energy consumption and low yield, to be solved. It is necessary to establish a physical and chemical mechanism model of the corresponding process and to reduce the energy consumption of the reaction process and improve the yield of the reaction product. The process needs to be optimized and designed and the optimal design parameters and operation plan of the process also need to be determined.

Many scholars have studied the two-step reactions of short-chain hydrocarbon polymerization-catalyzed synthetic fuel from the perspectives of classical thermodynamics and pure reaction kinetics. For CHLO, Li et al. [3] prepared a class of ZnZrO/SAPO tandem catalysts with ZnO-ZrO_2_ solid solution and Zn-modified SAPO-34 zeolite, which achieved 80–90% of the carbon olefin selectivity. Wu et al. [4] used porous graphene-coated Fe-K as a catalyst to study the reaction of CHLO. For olefin oligomerization, Albahily et al. [5] added divalent Cr to the ethylene oligomerization system and studied the effect of divalent Cr on the ethylene oligomerization system. Behr et al. [6] achieved high selectivity of isobutene dimerization and trimerization to prepare linear-ligated products using nickel catalysts.

In order to realize the engineering application of this technology, it is necessary to further combine the chemical reaction with the chemical process and analyze the optimal operation parameters of different components to achieve the optimal process performance. As a coupling discipline that includes heat transfer, fluid mechanics, chemical reaction kinetics, and other disciplines, finite-time thermodynamics [7,8,9,10,11,12,13,14,15] can be effectively used to solve the coupling process problems involving multiple processes. The short-chain hydrocarbon polymerization-catalyzed synthetic fuel process is a coupling process of heat transfer, fluid flow, and chemical reaction, so finite-time thermodynamics can be effectively used for its optimization research. In terms of the finite-time thermodynamic optimization of the chemical process, Chen et al. [16] and Zhang et al. [17] considered the irreversibility losses of finite-rate heat transfer, fluid flow, and chemical reaction, and optimized the one-dimensional CHLO reactor model with the goals of a minimum entropy generation rate (MEGR) and maximum yield. Li et al. [18,19] took the MEGR as the optimization goal and modeled and optimized the molten-salt-heating SMR reactor and membrane reactor. Kong et al. [20,21,22] took the conversion rate of hydrogen iodide, the maximum hydrogen recovery rate, and the MEGR as the optimization goals and optimized the high temperature helium-heating hydrogen iodide decomposition reactor and iodide hydrogen decomposition membrane reactor. Gunes et al. [23] analyzed the relation between heat transfer rate and entropy generation for single-pressure and dual-pressure waste heat recovery boilers, and showed that bigger boilers are both more efficient for heat transfer and have less entropy generation per heat transfer. Karakurt and Gunes [24] presented a useful combination of a mean cycle irreversibility for thermodynamically optimizing the Rankine cycle using the MCI as the currently proposed criterion and showed that the criterion produced more beneficial information to designers and engineers in terms of exergy destruction for designing more environmentally friendly and smaller thermal systems. Other researchers [18,19,20,21,22,23,24] considered the influence of the external heat source working medium, optimized the two-dimensional reactor model considering the axial factor and extended the research object from the traditional tubular reactor to the membrane reactor. However, only considering the single component of the reactor has certain limitations for improving the whole chemical process because the local optimum is not equivalent to the global optimum. Therefore, Zhao et al. [25] further extended the research object from a single reactor to a more complete reaction process, including mixers, compressors, heat exchangers, and chemical reactors, and optimized the reverse water gas shift reaction process with the goal of MEGR. Yu et al. [26] established a sub-process model of olefin oligomerization including mixer, compressor, heat exchanger, and chemical reactor, and carried out dual-objective optimization of the process with the goal of MEGR and maximum yield.

In this paper, on the basis of Ref. [26], the influence of olefin oligomerization sub-process is introduced. An overall process model for short-chain hydrocarbon polymerization-catalyzed synthetic fuel including two sub-processes of CHLO and olefin oligomerization were established and a separator was used to connect the two sub-processes. On the basis of the overall process model, the total process inlet component mole fraction, the total inlet molar flow rate, and the compressor outlet pressure in the two sub-processes were taken as the optimization variables and the MEGR of the total process was taken as the optimization objective. The influence of each optimization variable on the performance of the total process was analyzed and, finally, the performance optimization of short-chain hydrocarbon polymerization-catalyzed synthetic fuel technology was realized.

## 2. Physical Model of Short-Chain Hydrocarbon Polymerization-Catalyzed Synthetic Fuel Process

The short-chain hydrocarbon polymerization-catalyzed synthetic fuel process includes two sub-processes of CHLO and olefin oligomerization. Each sub-process includes a mixer, a compressor, a heat exchanger, and a reactor, and the two sub-processes are separated by a separator. Figure 1 shows the schematic diagram of a catalyzed synthetic fuel process including two sub-processes of CHLO and olefin oligomerization.

As can be seen in Figure 1, the raw material gas required for the reaction is usually stored in a compressed state in a cylinder in industry and energy is also required to compress the gas from a low-pressure state to a high-pressure state. In order to fully describe the performance of the CHLO sub-process, we took the ambient temperature (298.15 K) and ambient pressure (0.101 MPa) as the starting point of the process. The raw gas enters the mixer for mixing under ambient conditions and then enters the compressor for adiabatic pressurization. The pressure of the mass is raised to the pressure required for the reaction. Subsequently, the working fluid is passed into the heat exchanger to increase the temperature at the same pressure so that the temperature rises to the temperature required at the inlet of the reactor. The working fluid enters into the CHLO reactor for reaction, the obtained product subsequently enters the separator, and the C_2_H_4_, C_4_H_8_ and newly added N_2_ in the obtained product enter the compressor and heat exchanger to make the working fluid reach the temperature and pressure required for the reaction. Finally, the working fluid is passed into the olefin oligomerization reactor for reaction. Considering that the yield of CHLO is low and the value is too small, multiple CHLO sub-processes are selected to be connected with one olefin oligomerization sub-process.

According to Ref. [26], the physical model of each unit component is used in this paper. Among them, the entropy generation rate (EGR) of the mixer can be obtained by calculating the change in entropy at the outlet of the mixer and that at the inlet:(1)$$\Delta S_{M}=-R\sum_{k}n_{k}\ln y_{k}$$
where R is the molar gas constant, $n_{k}$ is the amount of substance of each component, and $y_{k}$ is the mole fraction. For the CHLO, k = CO_2_, H_2_, H_2_O, CO, C_10_H_20_, C_2_H_4_, C_4_H_8_. For the olefin oligomerization, k = C_2_H_4_, C_4_H_8_, C_10_H_20_, N_2_.

The EGR of the compressor can be obtained by calculating the change in entropy at the outlet of the compressor and that at the inlet:(2)$$\Delta S_{C}=F_{T}\sum_{k}y_{k}\int_{T_{\mathrm{out}}}^{T_{\mathrm{out}}^{\mathrm{isen}}}\frac{C_{p,k}}{T}dT$$
where $F_{T}$ is the molar flow rate, *T* is the working fluid temperature, and $C_{p,k}$ is the constant pressure molar heat capacity of the substance. The composition of the CHLO reaction process is different from that of the olefin oligomerization reaction process. In addition, for CHLO, $F_{T}$ is the total inlet molar flow rate of the total process, while the $F_{T}$ for olefin oligomerization is determined by the CHLO sub-process.

The EGR of the heat exchanger can be expressed as:(3)$$\Delta S_{H}=\int_{0}^{L}\pi d_{H,i}q_{H}(\frac{1}{T}-\frac{1}{T_{a}})\mathrm{dz}$$
where d is the inner diameter and $T_{a}$ is the temperatures of the heat source. $q_{H}$ is the heat flux, can be obtained by $q=U(1/T-1/T_{a})$, and U is the heat transfer coefficient. For the CHLO and olefin oligomerization reaction processes, they have different heat transfer coefficients and heat source temperatures outside of the heat exchanger tubes. 

The EGR of the reactor can be expressed as:(4)$$\Delta S_{R}=\int_{0}^{L_{R}}\pi d_{i}U_{R}{\left(1/T-1/T_{R,a}\right)}^{2}+A_{c}v_{m}[-\frac{1}{T}(\frac{dp}{dz})]+A_{c}\rho_{b}\sum_{j}r_{j}(\frac{\Delta_{r}G_{j}}{T})]\mathrm{dz}$$
where $v_{m}$ is the average flow rate, $A_{c}$ is the cross-sectional area, and $\rho_{b}$ is the bulk density of catalytic bed. There are three items in the integrand function of Equation (4). The first item is the EGR in the heat exchange process, the second item is the EGR in the flow process, and the third item is the EGR in the chemical reaction process. For two different reactions, the calculation results will be different due to different factors, such as reaction process types and heat source temperatures.

For the CHLO, the chemical reaction equation is:(5)$${\mathrm{CO}}_{2}{+H}_{2}\Leftrightarrow {\mathrm{CO}+H}_{2}{O\;\Delta }_{r}H_{1}>0$$
(6)$${\mathrm{CO}+H}_{2}\Leftrightarrow \frac{1}{6}C_{3}H_{6}+\frac{1}{2}H_{2}O\;\Delta_{r}H_{2}<0$$
(7)$${\mathrm{CO}+H}_{2}\Leftrightarrow \frac{1}{20}C_{10}H_{20}+\frac{1}{2}H_{2}O\;\Delta_{r}H_{3}<0$$
(8)$${\mathrm{CO}+H}_{2}\Leftrightarrow \frac{1}{3}{\mathrm{CH}}_{4}+\frac{1}{3}H_{2}O\;\Delta_{r}H_{4}<0$$

The reaction rate expression for each reaction is:(9)$$r_{1}=k_{1}(p_{CO_{2}}p_{H_{2}}-p_{CO}p_{H_{2}O}/K_{1})$$
(10)$$r_{2}=k_{2}(p_{CO}^{1/2}p_{H_{2}}-p_{C_{3}H_{6}}^{1/6}p_{H_{2}O}^{1/2}/K_{2})$$
(11)$$r_{3}=k_{3}(p_{CO}^{1/2}p_{H_{2}}-p_{C_{10}H_{20}}^{1/20}p_{H_{2}O}^{1/2}/K_{3})$$
(12)$$r_{4}=k_{4}(p_{CO}^{1/3}p_{H_{2}}-p_{{\mathrm{CH}}_{4}}^{1/3}p_{H_{2}O}^{1/3}/K_{4})$$
where p is the partial pressure, $k_{i}$ is the reaction rate constant, and $K_{i}$ is the reaction equilibrium constant.

For olefin oligomerization, the chemical reaction equation is:(13)$$C_{2}H_{4}\Leftrightarrow \frac{1}{2}C_{4}H_{8}{\;\Delta }_{r}H_{1}<0$$
(14)$$C_{2}H_{4}\Leftrightarrow \frac{1}{5}C_{10}H_{20}\;\Delta_{r}H_{2}<0$$

The reaction rate expression for each reaction is:(15)$$r_{1}=k_{1}(p_{C_{2}H_{4}}-p_{C_{4}H_{8}}^{1/2}/K_{1})$$
(16)$$r_{2}=k_{2}(p_{C_{2}H_{4}}-p_{C_{10}H_{20}}^{1/5}/K_{2})$$

For the separator, assuming that the separator process is isothermal and isobaric, no reaction occurs during the process and the separation efficiency is 0.15 [27]; the actual power consumption can be obtained from the system power consumption in the reversible situation:(17)$$W_{\mathrm{rev}}=-RT\left(\sum_{j=1}^{e}n_{\mathrm{Fj}}\ln x_{\mathrm{Fj}}-\sum_{j=1}^{m}n_{\mathrm{Qj}}\ln x_{\mathrm{Qj}}\right)$$
where *W*_rev_ is the reversible power consumption of the separator; *n*_Fj_ and *n*_Qj_ are the amounts of feed and product substances in the separator, respectively; *x*_Fj_ and *x*_Qj_ are the mole fractions of the feed and product, respectively; and e and m represent the component quantities of the feed and product.

The separation efficiency is the ratio of the reversible power consumption of the separator to the actual power consumption:(18)$$\eta =\frac{W_{\mathrm{rev}}}{W_{\mathrm{act}}}$$
In Equation (18), *W*_act_ is the actual power consumption of the separator.

The EGR of the separator can be calculated by $(W_{\mathrm{act}}-W_{\mathrm{rev}})/T$, i.e.,
(19)$$\Delta S_{S}=-\frac{W_{\mathrm{act}}}{T}-R\left(\sum_{j=1}^{e}n_{\mathrm{Fj}}\ln x_{\mathrm{Fj}}-\sum_{j=1}^{m}n_{\mathrm{Qj}}\ln x_{\mathrm{Qj}}\right)$$

## 3. Optimization Methods

### 3.1. Problem Description

The optimal performance problem is to solve the extremum of the objective function of the total short-chain hydrocarbon polymerization-catalyzed synthetic fuel process flow under given constraints and to obtain the optimal parameter values when the objective function achieves the extremum. The research results can provide theoretical guidance for the selection of operation parameters of the general short-chain hydrocarbon polymerization-catalyzed synthesis of fuel process.

The optimal performance problem is to solve the MEGR of short-chain hydrocarbon polymerization-catalyzed synthetic fuel with the constant temperature of the heat source outside the tubes of heat exchangers and reactors. Under the constraint of the given C_10_H_20_ yield, the total molar flow rate at the inlet of the mixer for the CHLO sub-process, the inlet mole fraction of each component, and the compressor outlet pressure are chosen as the optimization variables, the MEGR of the sub-process and the values of the corresponding optimization variables can be determined and the influence of each optimization parameter on the total EGR can also be analyzed. According to Ref. [26], the optimal ranges of $p_{\mathrm{in}}^{C}$ and $p_{\mathrm{in}}^{O}$ are set to be 2–4.5 MPa and the optimal range of *F*_T,in_ is 0.15–2 mol/s. The optimization objective of the problem is to determine the minimum value of Equation (19) and the corresponding constraints of the optimization problem are listed in Table 1.

### 3.2. Optimization Problem Classification

In this work, we gradually released the constraints of the optimization variables *F*_T,in_, *y*_k,in_, $p_{\mathrm{in}}^{C}$, and $p_{\mathrm{in}}^{O}$ on the basis of the given reference parameters of the total inlet molar flow rate *F*_T,in_, the inlet mole fraction *y*_k,in_ of each component, and the compressor inlet and outlet pressures, i.e., $p_{\mathrm{in}}^{C}$ and $p_{\mathrm{in}}^{O}$, to achieve process optimization. Specifically, the optimization problem is divided into the following four steps:

(1) Solve the reference process performance in which the heat source outside of the heat exchanger and the reactor is constant and the values of the optimization variables are given. Among them, the temperatures $T_{a1}^{C}$ and $T_{a2}^{C}$ of the heat source outside the heat exchanger and the reactor tube for the CHLO sub-process are constant and equal to 573 K and the outlet pressure $p_{\mathrm{in}}^{C}$ of the compressor is 3 MPa; in the olefin oligomerization sub-process, the heat source temperature $T_{a1}^{O}$ outside the tube of the heat exchanger is constant and equal to 644 K, the heat source temperature $T_{a2}^{O}$ of the reactor is constant and equal to 637 K, and the outlet pressure $p_{\mathrm{in}}^{O}$ of the compressor is 3 MPa. The corresponding calculation results for this process are denoted as Ref in the following and the calculation results of the reference process are used as the initial values and the benchmark for the subsequent process optimization.

(2) Release the constraint of *y*_k,in_ and use it as an optimization parameter to optimize the overall process. Among them, $p_{\mathrm{in}}^{C}$, $p_{\mathrm{in}}^{O}$, and *F*_T,in_ are the same as those for the reference process. The corresponding calculation results are denoted as Process 1 in the following.

(3) On the basis of Process 1, the constraints of $p_{\mathrm{in}}^{C}$ and $p_{\mathrm{in}}^{O}$ are further released and added as optimization variables to optimize the overall process, where *F*_T,in_ is the same as the reference flow. The corresponding calculation results are denoted as Process 2 in the following.

(4) On the basis of Process 2, the constraint of *F*_T,in_ is further released and *F*_T,in_ is added as an optimization variable to optimize the overall process. The corresponding calculation results are denoted as Process 3 in the following.

### 3.3. Method of Solving

The nonlinear programming numerical method is used to solve the optimal performance of the short-chain hydrocarbon polymerization-catalyzed synthetic fuel process. The objective function of this problem is the total entropy yield of the short-chain hydrocarbon polymerization-catalyzed synthetic fuel process. Among them, the inlet state of the olefin oligomerization sub-process is determined by the outlet state of the CHLO reactor. By writing the MATLAB program m file for calculating the EGR of a single component, with the C_10_H_20_ yield as the constraint condition, based on the “fmincon” optimization toolbox in MATLAB, 9 m files were called for solving the unit EGR to obtain the objective function file and the entire process was finally optimized.

The reference process was constructed with reference to the actual engineering experience [28,29,30]. Table 2 shows the mole fraction of each component at the inlet of the process and Table 3 shows the key parameters of the heat exchanger and the reactor.

## 4. Numerical Example of Univariate Analysis

The reference process was calculated by the “ode45” solver and the value of $\Delta F_{C_{10}H_{20}}$ is 0.1133 mol/s. The number of micro-elements for discretization of the differential equation was set to 201 and the optimization problem was solved.

Table 4 shows the comparison of optimization variables of the processes for four different cases. Table 5 shows the comparison of the EGRs of four cases of each sub-process and the separator for four different cases. It can be seen that the EGR of the CHLO sub-process and separator is the main component of the EGR of the overall process.

Table 6 shows the EGRs of each component in the single CHLO sub-process under four different cases. According to Equations (1) and (2), it can be seen that the increase in the mole fraction of H_2_ and CO is beneficial to the mixer and compressor optimization. Compared with Process 1, Process 2 reduces the compressor outlet pressure. According to Equation (2), the reduction in the compressor outlet pressure is beneficial to the compressor optimization and the total EGR of the sub-process is reduced by 5.94%. Process 3 reduces the total inlet molar flow rate compared to Process 2. According to Equation (2), reducing the total molar flow rate at the inlet can reduce the EGR of the compressor. Compared with Process 2, the EGR of the CHLO sub-process in Process 3 is reduced by 20.26%.

Figure 2 shows the working fluid temperature curves in the CHLO reactors under four different cases. For Ref, since CO_2_ and H_2_ are the main components, the chemical reaction in Equation (5) is the main reaction in the reactor. Since the chemical reaction in Equation (5) is an endothermic reaction, the temperature of the working fluid drops in the initial stage. With the progress of the chemical reaction in Equation (5), CO begins to participate in the reaction and the chemical reactions in Equations (6)–(8) begin to become exothermic so that the reaction temperature gradually increases. For Process 1, by reducing the content of CO_2_ and increasing the content of CO, the degree of the chemical reaction in Equation (5) is reduced so there is no longer a decrease in the initial temperature and the EGR of the chemical reaction process is also reduced. The EGR of the reactor is also reduced. Process 2 reduces the pressure of the reaction process, which makes the balance of the chemical reaction move to the reverse reaction direction, prolongs the time required for the reaction, and also reduces the intensity of the reaction and the EGR of the reaction process is also reduced. Process 3 reduces the total molar flow rate at the inlet of the process and increases the proportion of CO in the working fluid, which reduces the intensity of the chemical reaction to ensure the yield, and reduces the EGR of the reaction process.

Table 7 shows the mole fraction of each component at the outlet of the CHLO sub-process under four different cases. In the four cases, the total molar flow rates at the outlet of the CHLO sub-process are 0.1622 mol/s, 0.1575 mol/s, 0.1600 mol/s, and 0.0604 mol/s, respectively. For the separator, it can be seen in Equation (11) that its EGR is related to the component mole fraction as well as the total molar flow rate. When the molar fraction distribution of each component is closer to the pure substance, that is, when there are more molar fractions of the main components, the EGR of the mixing process becomes lower. In the three cases of Ref, Process 1, and Process 2, H_2_ is the main component at the outlet of the CHLO sub-process. By increasing the proportion of H_2_, the composition of the components is purer, and the EGR of the separator is reduced. For Process 3, the total molar flow rate at the outlet of the CHLO sub-process decreases. 

## 5. Conclusions

In this work, a short-chain hydrocarbon polymerization-catalyzed synthetic fuel process model including CHLO sub-process, olefin oligomerization sub-process, and separator was established. Under the condition that both the total C_10_H_20_ yield and the temperature of the external heat source are given, the MEGRs of the process were determined for three different cases and are compared with the performance of the reference process herein. The results are as follows:

1. The total EGR is reduced by 29.60%, 30.97%, and 48.81%, respectively, when the overall process is optimized by gradually adding the component mole fraction, pressure, and total molar flow rate as optimization variables.

2. The EGRs of the CHLO sub-process and the separator are the main components of the EGR of the whole process and the optimization effect is mainly to reduce the EGRs of these parts.

3. Compared to the case that only the inlet pressure is chosen as an optimization variable, the optimization effects are more obvious when the molar fraction of components and the total molar flow rate are further chosen as optimization variables.

In addition to the EGR performance index, ecological efficiency was often used as a performance index to evaluate the thermodynamic system [31,32,33,34]. The methodology for environmental analysis using ecological efficiency was initially proposed by Cardu and Baica [31,32] for thermal power plants with steam cycles using coal as fuel. The ecological efficiency is an indicator that allows the performance of a thermoelectric power plant to be evaluated with respect to pollutant emissions by comparing hypothetically integrated pollutant emissions (CO_2_ equivalent emissions) to existing air quality standards [33,34]. Thus, the ecological efficiency or other approaches different from the EGR performance index will be our next research direction.

## Figures and Tables

**Figure 1 entropy-24-01658-f001:**
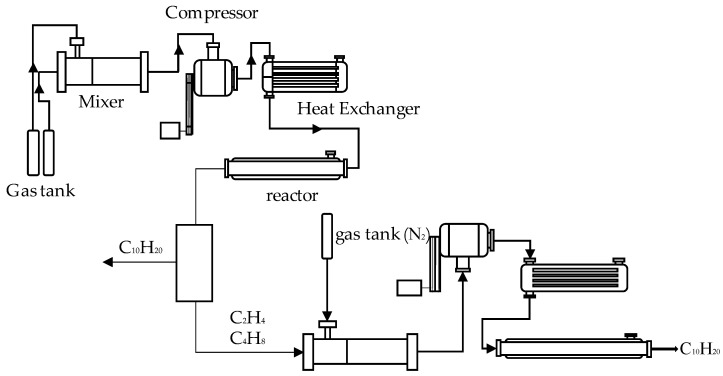
The schematic diagram of catalyzed synthetic fuel process including two sub-processes of CHLO and olefin oligomerization.

**Figure 2 entropy-24-01658-f002:**
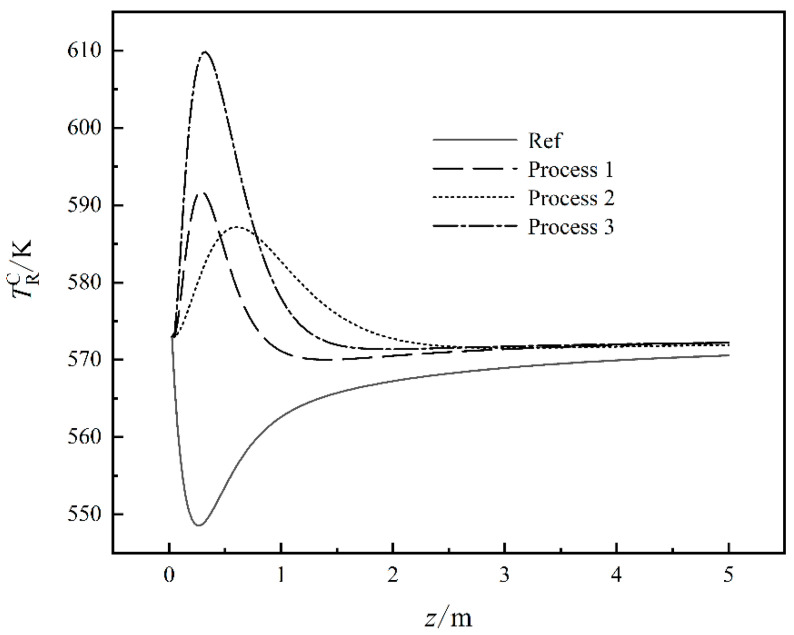
The working fluid temperature curves in the CHLO reactors under four different cases.

**Table 1 entropy-24-01658-t001:** Constraints of the optimization problem.

Constraint Name	Constraint Range
Component molar fraction constraints	$0\leq y_{k,\mathrm{in}}\leq 1$
Compressor inlet pressure constraint	$2\;\mathrm{MPa}\leq P_{\mathrm{in}}^{C}\leq 4.5\;\mathrm{MPa}$
Compressor outlet pressure constraints	$2\;\mathrm{MPa}\leq P_{\mathrm{in}}^{O}\leq 4.5\;\mathrm{MPa}$
total inlet molar flow rate constraint	$0.15\mathrm{mol}/s\leq F_{T,\mathrm{in}}\leq 2\mathrm{mol}/s$

**Table 2 entropy-24-01658-t002:** The mole fraction of each component at the inlet of the process [25,26].

**Component**	${\mathrm{CO}}_{2}$	$H_{2}$	CO	$C_{3}H_{6}$	$C_{10}H_{20}$	${\mathrm{CH}}_{4}$	$H_{2}O$
**Mole** **Fraction**	0.25	0.745	0.001	0.001	0.001	0.001	0.001

**Table 3 entropy-24-01658-t003:** The key parameters of the heat exchanger and the reactor [25,26].

**Parameters**	Heat transfer coefficient	Inner diameter	Outer diameter	length
**Symbol**	$U_{H}$ $,\;U_{R}$	$d_{i}$	$d_{o}$	$L_{1}$
**Value**	$1.7\times 10^{7}\;W\cdot K\cdot m^{-2}$	0.08 m	0.084 m	5 m

**Table 4 entropy-24-01658-t004:** Comparison of optimization variables of processes for four different cases.

Case	Ref	Process 1	Process 2	Process 3
$y_{{\mathrm{CO}}_{2},\mathrm{in}}$	0.25	$7.38\times 10^{-3}$	$1.6\times 10^{-4}$	$1.82\times 10^{-3}$
$y_{H_{2},\mathrm{in}}$	0.745	0.8715	0.8564	0.6770
$y_{\mathrm{CO},\mathrm{in}}$	0.001	0.1169	0.1401	0.3171
$y_{H_{2}O,\mathrm{in}}$	0.001	$1.23\times 10^{-3}$	$2.7\times 10^{-4}$	$1.07\times 10^{-3}$
$p_{\mathrm{in}}^{C}$	3 MPa	3 MPa	2 MPa	2 MPa
$p_{\mathrm{in}}^{O}$	3 MPa	3 MPa	2.89 MPa	2.48 MPa
$F_{T,\mathrm{in}}$	0.2 mol/s	0.2 mol/s	0.2 mol/s	0.1 mol/s

**Table 5 entropy-24-01658-t005:** Comparison of EGRs of four cases of each sub-process and separator for four different cases.

Case	Ref	Process 1	Process 2	Process 3
$\Delta S_{\mathrm{tot},C}$	435.74 W/K	414.87 W/K	390.24 W/K	311.18 W/K
$\Delta S_{S}$	922.43 W/K	542.28 W/K	547.29 W/K	383.24 W/K
$\Delta S_{\mathrm{tot},O}$	6.1841 W/K	3.7747 W/K	4.2696 W/K	3.9388 W/K
$\Delta S_{\mathrm{tot}}$	1364.35 W/K	960.925 W/K	941.799 W/K	698.359 W/K
Decrease ratio	——	29.60%	30.97%	48.81%

**Table 6 entropy-24-01658-t006:** The EGRs of components in the single CHLO sub-process under four different cases.

Case	Ref	Process 1	Process 2	Process 3
$\Delta S_{M,C}$	0.9984 W/K	0.7250 W/K	0.7191 W/K	0.5552 W/K
$\Delta S_{C,C}$	1.1516 W/K	1.0190 W/K	0.9259 W/K	0.4655 W/K
$\Delta S_{H,C}$	0.0231 W/K	0.0203 W/K	3.0540 × 10^−6^ W/K	1.5447 × 10^−6^ W/K
$\Delta S_{R,C}$	2.1843 W/K	2.3844 W/K	2.2574 W/K	2.0910 W/K

**Table 7 entropy-24-01658-t007:** The mole fraction of each component at the outlet of the CHLO sub-process under four different cases.

Case	CO_2_	H_2_	CO	C_3_H_6_	C_10_H_20_	CH_4_	H_2_O
Ref	0.1747	0.5148	0.0073	0.0167	0.0078	0.0164	0.2623
Process 1	0.0096	0.7879	0.0009	0.0188	0.0083	0.0259	0.1487
Process 2	0.0336	0.8138	0.0048	0.0185	0.0082	0.0173	0.1038
Process 3	0.1587	0.5206	0.0088	0.0490	0.0200	0.0365	0.2064

## Data Availability

The data that support the findings of this study are available from the corresponding author upon reasonable request.

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
