# Peer review of "Finite-Time Thermodynamic Modeling and Optimization of Short-Chain Hydrocarbon Polymerization-Catalyzed Synthetic Fuel Process"

_entropy, 2022, doi:10.3390/e24111658_

Round 1
Reviewer 1 Report (Previous Reviewer 1)
Dear Authors, please, read the file attached.
Best regards.

Author Response
None.
Reviewer 2 Report (New Reviewer)
thank you for your valuable effort. I added some suggestions which can be helpful for your future preparations. .
The introduction part can be improved with relevant studies such as shorturl.at/jCGUY, shorturl.at/ceqA5, etc.
Four different cases were mentioned in the results but I couldn't find any detail about the cases. Please, make the difference clear between cases.
As an important comment, adding ecological efficiency or different approaches for EGR, such as the above links, analyses have to make the paper stronger.
Definitions of tables and figures and also the Conclusion part must be detailed.
Please share the constraints as a table.
Fig. 1 can be well described and named with the subsystem.
Tables 2 & 3 can be combined.
I am not sure if equation number 20 is an equation.
Round 2
Reviewer 2 Report (New Reviewer)
thank you for your revisions.
This manuscript is a resubmission of an earlier submission. The following is a list of the peer review reports and author responses from that submission.
Round 1
Reviewer 1 Report
Please, see thefile attached.

Reviewer 2 Report
Thermodynamic optimization of processes has been extensively studied by S. Kjelstrup et al. and others. In the present work, no novel contributions can be considered a complete article; there is neither analysis of the thermodynamic formalism of irreversible processes nor of the thermodynamic rigor of optimization. The authors present a collection of formulas, all known, and at the end, the article ends abruptly with a table of results; this shows that in this work, a numerical method is mainly being applied to solve a set of formulas.